# I struggle with your fidgeting: A qualitative study of the personal and social impacts of misokinesia

**Sumeet M. Jaswal** [1]*, **Drake Levere**[2], **Todd C. Handy**[1]

1 Department of Psychology, University of British Columbia, Vancouver, British Columbia, Canada,
2 Blizzard Entertainment, Irvine, California, United States of America

* sumeet@psych.ubc.ca

## Abstract

Misokinesia, the reduced tolerance to others' repetitive bodily movements, impacts individuals' personal, social, and professional lives. The present study aimed to elucidate the factors contributing to Misokinesia Sensitivity (MKS) by exploring the subjective experiences of affected individuals, thereby guiding future empirical research and informing clinical awareness. Using a qualitative approach, we conducted semi-structured interviews with 21 participants from an online support group on Facebook. Data were analyzed through thematic analysis to uncover patterns and themes in their lived experiences. The thematic analysis revealed three main themes: internal cognitive and affective impacts, external social impacts, and pragmatic factors related to MKS triggers and responses. These findings provide a foundational understanding of MKS, highlighting its significant personal and social consequences and suggesting areas for targeted interventions. The insights gained aim to enhance clinical recognition and support the development of effective management strategies for those affected by MKS.

## Introduction

Misokinesia––or the 'hatred of movements'––is a condition characterized by a reduced tolerance to the sight of someone else's repetitive bodily movements, such as fidgeting with a hand or foot [1]. When confronted with such stimuli, those with misokinesia sensitivity (MKS) report an array of aversive responses, including strong emotional, physiological, and/or behavioural reactions that, for many, impact their personal, social, and professional life [2]. Yet what is the cause or underlying basis for this social-perceptual challenge? To our knowledge, only two empirical studies to date have addressed the question. One was designed to assess whether those with MKS show heightened attentional-orienting responses to sudden visual movements [2], and the other examined whether MKS is associated with heightened affective reactivity to social-visual stimuli [3]. However, no such relationships were found. As such, the underlying cause for MKS remains unknown.

Towards trying to better understand what factors may contribute to MKS, we thus performed the following qualitative, interview-based study with three related purposes of research in mind. First, our hope was that by systematically exploring the lived, subjective experience of

**Funding:** We received funding from Natural Sciences and Engineering Research Council of Canada (NSERC) for our study. The funders had no role in study design, data collection and analysis, decision to publish, or preparation of the manuscript.

**Competing interests:** The authors have declared that no competing interests exist.

those with MKS, it could then help inform the next generation of objective, empirical studies aiming to identify the functional or mechanistic basis for MKS. As such, we were adopting a qualitative approach previously taken to better understand the underlying contributors for misophonia [1, e.g., 4], the more widely-studied auditory analog to misokinesia. Second, MKS is highly prevalent within the general North American population, with approximately one-third of individuals sampled self-reporting some level of MKS [2], yet it remains a relatively unknown condition. In better understanding the lived experiences of those with MKS, we thus hoped to build a foundation for better informing the clinical/medical community about this widespread social-perceptual challenge. Finally, the present work focused on analyzing the perpetuating factors that contribute to the challenges associated with misokinesia. This idea is part of the 4P model proposed by Bolton [5], which categorizes case formulations in clinical disorders as predisposing, precipitating, perpetuating, and protecting factors. It was our hope that with increased insight into both the possible contributors to MKS and how MKS impacts peoples' daily lives, this could then also help inform on the development of targeted intervention strategies for mitigating the range of challenges that come with MKS.

Our interview questions as listed in Table 1 were inspired by Edelstein et al.'s [4] qualitative study of misophonia, and were designed to explore and inform on four core aspects of MKS as per our trio of study goals: (1) how it impacts the daily life and routines of those with MKS; (2) what specific types of seen movements are associated with MKS; (3) what kinds of reactions and/or responses are evoked when experiencing MKS; and (4) what sorts of coping mechanisms are employed to help mitigate the impacts of MKS, if at all. In deciding who to specifically recruit for interviews, there is no current or accepted diagnostic criteria for misokinesia as a clinical condition. Thus, as described below, we adopted a recruitment strategy that targeted individuals who had self-selected to join an on-line misokinesia support group. Following our set of interviews, we then used qualitative thematic analysis [6] to interrogate the data and address our study goals.

**Table 1. Interview questions regarding participants' misokinesia sensitivity.**

| | Interview Questions |
|---|---|
| 1 | What aspects of your life do you feel have been adversely impacted by misokinesia? |
| 2 | Can you describe what specific visual movements bother you the most? |
| 3 | Are there people or places you routinely avoid because of visuals that are likely to be encountered? |
| 4 | How have people reacted to you when you have told people about your symptoms? |
| 5 | What symptoms do you feel when you are being bothered by movement? |
| 6 | Have there been visual movements that previously bothered you, that you have now grown accustom to? |
| 7 | Generally, after experiencing a problem movement–how long does it take for you to "get past it" or "let it go"? |
| 8 | When a movement bothers you, how do you cope with it? i.e., do you distract yourself with another memory, mimic it, ignore it, or remove yourself from the situation, etc.? |
| 9 | What coping mechanisms do you use for dealing with your misokinesia? |
| 10 | Are there other non-fidget things that bother you? i.e., traveling in a bus while facing backwards, driving behind a car with rear wipers going when not raining, etc. |
| 11 | Have you noticed any animated or cartoon fidgeting that bothers you? Or is it specifically human fidgeting? |
| 12 | Have you noticed if babies that are fidgeting that bothers you? Or is it specifically human fidgeting? |
| 13 | Have you noticed animal fidgeting that bothers you? Or is it specifically human fidgeting? |
| 14 | Do you notice a difference between when you look at something fidgeting in your periphery vs looking straight at someone fidgeting, in terms of how much it bothers you? |
| 15 | Some people have mentioned that if they are near a "problem fidgeter" and they look at their hand or foot (even if it's not fidgeting yet), it brings them a great deal of anxiety–have you experienced anything similar? They say that it brings them anxiety because they do not know when the person may start fidgeting again. |

## Methods

All procedures, methods, and materials were approved by the UBC Behavioural Research Ethics Board (BREB). Participants were recruited from the "Misophonia and Misokinesia Support Group" on Facebook, a pre-existing group that gave us permission to advertise our study. We posted a recruitment flyer on April 6, 2020, which invited interested individuals "bothered by fidgeting" to participate in a 30-minute interview. When initial interest was expressed (via posting an on-line comment/request on our poster), we contacted the individual via Facebook private messaging to obtain their email address and send them further information and a formal interview invitation. For those ultimately agreeing, an online video meeting was scheduled, with all interviews in the study taking place via Zoom between April 9, 2020, and July 31, 2020. We had no further inclusion/exclusion criteria. Study recruitment continued from April 6, 2020 until interest in the study waned and then, in August, 2020, the group administrator declined our request to re-post the recruitment flyer.

A total of 21 individuals participated–– 19 females and 2 male participants, ranging in age from 18 to 64 (see Table 2 for further demographics). All provided informed consent prior to participation, self-reported as fluent in English, and received a $10 Amazon.com e-gift certificate for their time. Each interview lasted approximately 30 minutes, and was semi-structured (i.e., we asked the 15 questions in Table 1, and then followed-up as necessary as each question was idiosyncratically answered). Because we were denied ethics approval to record the interview sessions, all given answers were typed into a document as each interview unfolded, with one exception––one participant reported a high sensitivity to the sound of computer typing, and so for that participant responses were written by hand. Following completion of the interviews, participants were then asked to complete three questionnaires accessed on-line via a provided link: (1) the misokinesia assessment questionnaire [MkAQ; 2] 21-item self-report instrument that indexes the degree of one's MKS, (2) the misophonia assessment questionnaire [MpAQ; 7], a 21-itel self-report instrument that appraises the degree to which an individual experiences negative thoughts, feelings, and emotions regarding misophonic sounds, (3) a basic demographics questionnaire, and (4) the state and trait anxiety inventory [STAI; 8], which was included for use in an unrelated study. As the study was conducted during the

**Table 2. Demographic information for the interviewed participants.**

| Demographics | Frequency (percent) |
|---|---|
| Total Participants | 21 |
| Age | 18-24–5 (23.81%)<br>25-34–10 (47.62%)<br>35-44–4 (19.05%)<br>45-54–1 (4.76%)<br>55-64–1 (4.76%) |
| Sex | Females– 19 (90.48%)<br>Males– 2 (9.52%) |
| Race/ethnicity | White or Caucasian– 13 (61.90%)<br>South Asian– 5 (23.81%)<br>Hispanic– 1 (4.76%)<br>Pacific Islander– 1 (4.76%)<br>Aboriginal Australian– 1 (4.76%) |
| Country of Residence | USA– 12 (57.14%)<br>Canada– 6 (28.57%)<br>Mexico– 1 (4.76%)<br>Nepal– 1 (4.76%)<br>Australia– 1 (4.76%) |

COVID-19 pandemic [9], participants were only recruited and interviewed online via Zoom (Zoom Video Communications, Inc.), which became a popular software during the pandemic that all participants were aware of prior to the research study.

After preliminary analysis of the data as described below, we determined that five questions could benefit from further elaboration from the participants. Accordingly, the UBC BREB gave us approval to recontact our original 21 study participants and invite them to fill out an on-line questionnaire asking for more detailed responses to a set of five questions, which were: (1) Can you describe what aspects of your life you feel have been adversely impacted by misokinesia?; (2) What are the people or places you routinely avoid because of the misokinesic movement that you will likely encounter?; (3) How have people reacted to you when you have told people about your misokinesia?; (4) What symptoms do you feel when you are bothered by misokinesic movement?; and (5) How do you cope with your misokinesia? The email invitation was sent to all 21 participants on October 27, 2021; three ultimately agreed to provide this further detail, and it was then included in the thematic analysis.

This study employs qualitative thematic analysis as outlined by Braun and Clarke [6]. Thematic analysis is a method for identifying, analyzing, and reporting patterns (themes) within data. The study was performed from a constructivist/interpretivist point of view which believes that individuals or groups construct reality based on interactions with the social environment [10]. We employed this paradigm since it emphasizes understanding phenomena from the perspective of those who experience them, which aligns with the study's objective to explore the personal and subjective experiences of individuals with MKS.

The first author (SMJ) conducted all interviews from an online support group that they found on Facebook. SMJ was not a part of this support group prior to beginning the recruitment process for this study, nor was she known to the participants of this research prior to undertaking the study. She made sure to write down what the participants were saying in the interviews, checked that she had written the correct quotes by reading them back to the participants, and made a conscious effort not to assume that they "knew" what the participants "meant" to say; rather she explicitly noted what the participants had told her in the interviews.

For thematic analysis, we interrogated the interview data by following the guidelines for qualitative thematic analysis as proposed by Braun and Clarke (2006), which breaks down the analysis into six core phases. In Phase 1, the written answers for each participant were read through multiple times by one of the authors (SMJ) in order to gain general familiarity with the material, and then initial ideas for coding patterns in the data––or how to group salient or important/relevant interview extracts together––were identified. In Phase 2, SMJ established a beginning set of codes and collated interview extracts from across the data set that aligned with each code. In Phase 3, SMJ then generated an initial set of themes by grouping related codes (or interview extracts) together. In Phase 4, SMJ reviewed whether these initial themes appropriately aligned with their associated coded extracts, adjusted the theme-code mapping as appropriate, and then generated a thematic "map" of the analysis. In line with Braun and Clarke [6], this involved double-checking and further evaluating that the coded extracts placed within each initial theme made conceptual sense in generating a thematic map of the analysis. To ensure the validity of each theme and code during this phase, our criterion was that three or more participants had to have quotes––or interview extracts––speaking to that particular code or theme; data saturation was reached when all relevant quotes were categorized into themes/subthemes, and no new themes could be generated from the dataset. In Phase 5, all three authors considered/refined the thematic map, including the theme/sub-theme names and definitions, the codes subsumed under each theme, and the general narrative the analysis was supporting with respect to our study's goals. It was during this phase that we concluded more detailed information would be helpful to inform on our initial set of themes, leading to

our follow-up survey request in October, 2021 as reported above; extracts from the survey data set were then added to the appropriate themes/subthemes. Finally, in Phase 6, SMJ selected the specific interview extracts to include here, SMJ and TCH co-wrote the paper, and DL reviewed the paper.

The checklist Standards for Reporting Qualitative Research (SRQR) guided the reporting of this study [11]. The data necessary to reproduce the findings regarding the misokinesia (MkAQ), misophonia (MpAQ), and anxiety (STAI) scores reported in this manuscript are available in the supporting document (S1 Table).

## Results

### MkAQ scores

As an instrument for assessing MKS, the MkAQ asks a set of 21 questions that each concern some potential aspect of MKS; each of these questions can be answered by entering a value of 0 to 3, where the higher the value on that question, the higher the self-reported MKS (Jaswal et al., 2021). As such, an individual's total score on the MkAQ can range from 0 (or no reported sensitivity to any of the questions) to 63 (or maximum reported sensitivity on all 21 questions). In Fig 1 we plot the frequency distribution of the MkAQ scores in our sample, which, as expected given our recruitment from an on-line misokinesia support group, was somewhat negatively skewed towards higher levels of MKS (range: 4–63, M = 40.1, SD = 18.7).

### Misophonia

We examined whether there was an overlap between our analyses of misokinesia and misophonia, an auditory counterpart to misokinesia. Specifically, misophonia is defined by excessive emotional distress to specific triggering sounds such as chewing and lip-smacking (Dozier, 2015b), and individuals reporting high levels of misophonia often report high levels of misokinesia as well (e.g., Jaswal et al., 2021; Schröder et al., 2013). Consistent with this, in the current study, we found that participants' scores on the MpAQ positively correlated with their scores on the MkAQ ($r = 0.811$, $p < 0.001$; Fig 2).

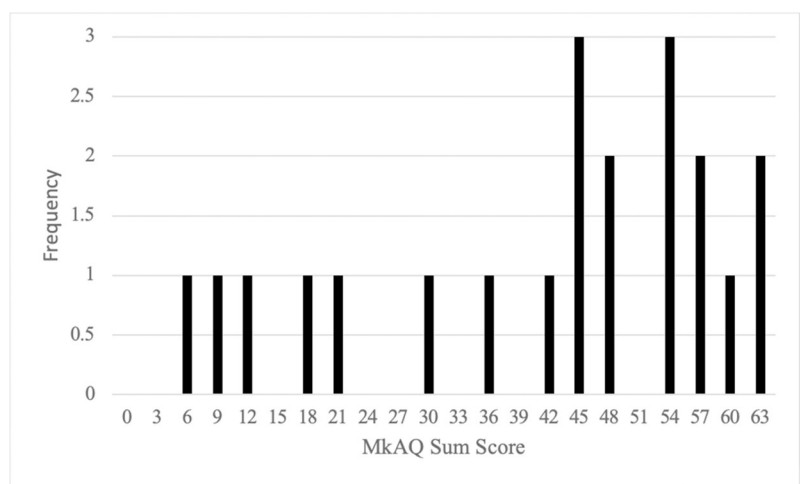

**Fig 1. Frequencies of MkAQ sum scores plotted for our participants.**

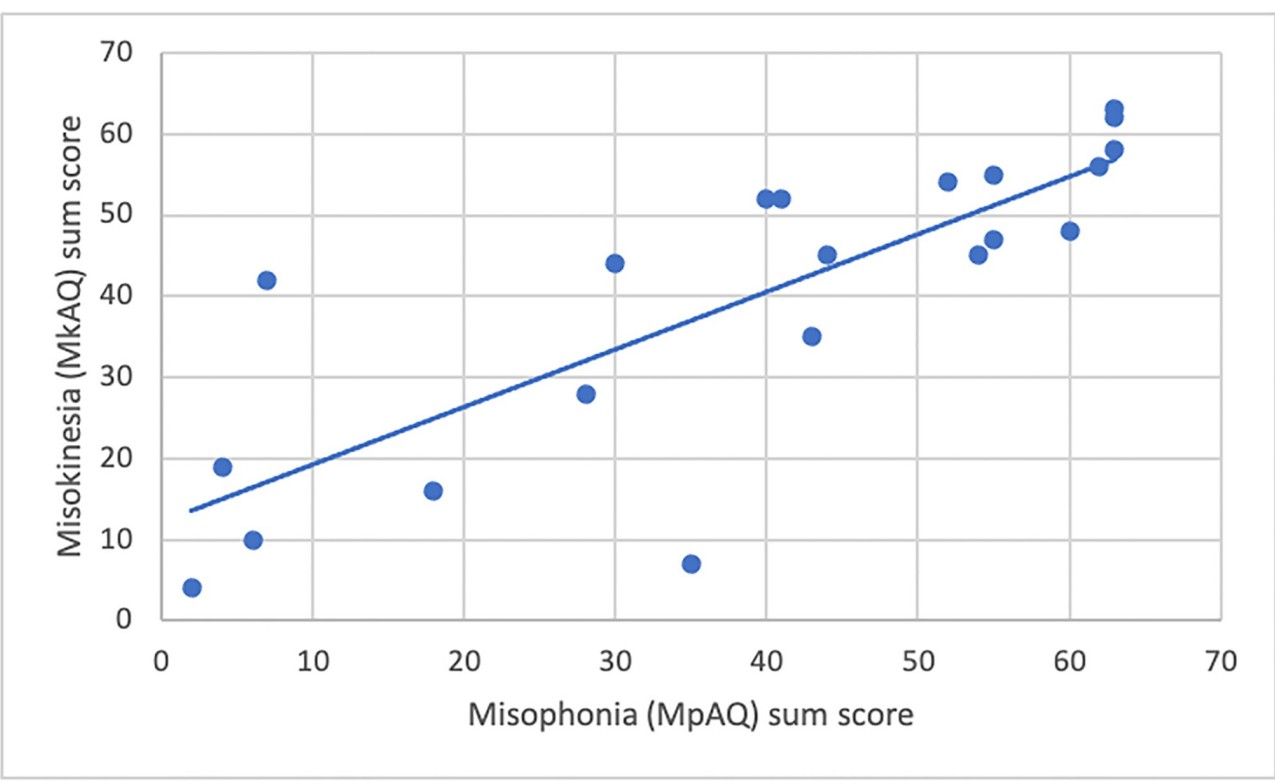

**Fig 2. Correlation of misokinesia (MkAQ) and misophonia (MpAQ) sum scores.**

### Anxiety

We also evaluated if participants' state or trait anxiety contributed to a predisposition for misokinesia and additional impairments. Consequently, we correlated the total scores on the MkAQ with their anxiety scores (either state or trait). The findings indicated that misokinesia scores did not correlate with state anxiety ($r = .308$, $p = .17$) or with trait anxiety ($r = .187$, $p = .429$).

Interviews were anonymized and each participant was given a code number at the time of initial interviews so they could input the code number into the questionnaires as they completed them. Then, when we re-contacted participants to participate in our study again in October 2021, we sent an email to all participants and asked participants to fill in their initial code number again.

### Thematic analysis

Our finalized thematic map is shown in Fig 3. Our analysis converged on three overarching themes that directly speak to––or reflect––our specific study goals of hoping to inform on (1) the possible underlying mechanistic basis for MKS, (2) the lived experiences of those with MKS, and (3) possible intervention strategies for treating MKS. The first theme we describe below concerns impacts of MKS that we categorized as specific to the individual's internal (or personal) cognitive, affective, and physiological experiences. The second theme we describe below concerns impacts of MKS that we categorized as specific to the individuals' external (or social) experiences. Finally, the third theme concerns the more pragmatic factors or parameters associated MKS, including the types of stimuli that do (and don't) evoke responses, the

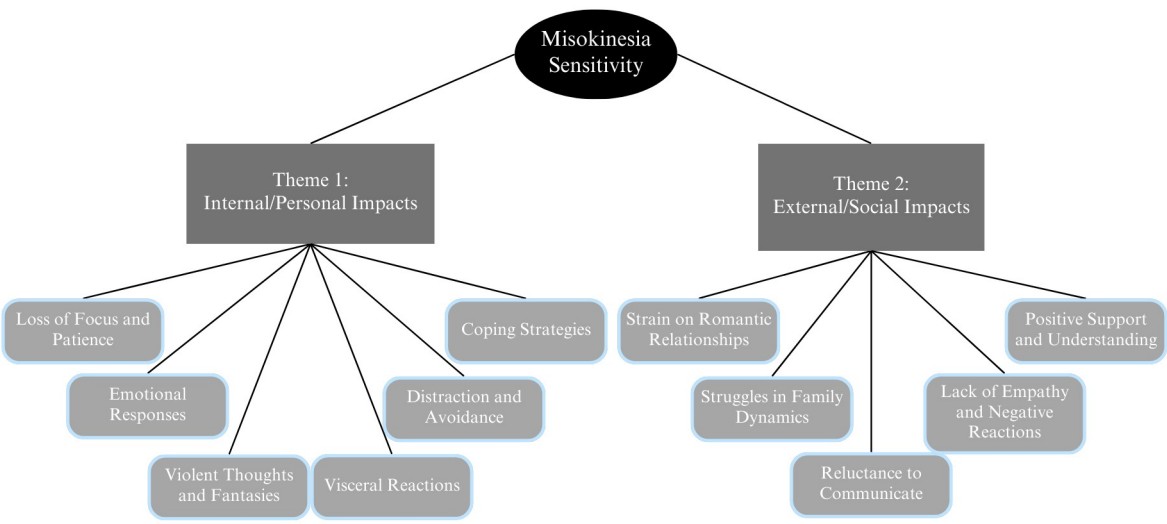

**Fig 3. Thematic map of misokinesia sensitivity (MKS).**

duration of evoked experiences, and changes in MKS sensitivity over the lifespan. Notably, the extracts quoted below also highlight how similar themes and subthemes can be seen within responses to different questions, qualitative overlap which served to cross-validate both the experiences reported by participants and our thematic coding of those reports.

**Theme 1: Internal/personal impacts.** *Subtheme 1.1*: *Loss of focus and patience*. Participants consistently reported a profound loss of concentration and patience when exposed to triggering movements or sounds, affecting their ability to stay present in various settings, including academic, professional, and social environments.

> *"My ability to focus is gone. I have no ability to ignore or disregard the sound or motion. I am unable to stay present."*

> *"For the most part, my ability to concentrate, patience, and sanity are all gone."*

> *"I would say my ability to be in a focused environment such as school or religious gatherings has been negatively affected."*

> *"If I am watching a movie, and someone starts fidgeting, I can no longer concentrate on the film. Instead, I exert all my attention on the annoyance of the fidget."*

> *"I experience a fixation, like a hyper-fixated on the noise or the motion. The sound gets louder in my head or the motion becomes the only thing I can focus on."*

> *"I'm not able to focus on anything else, and my whole body feels like it's in some sort of high alert situation. Like when you think you're being followed by someone that might want to hurt you."*

*Subtheme 1.2*: *Emotional responses*. Emotional reactions to triggers ranged from general annoyance to extreme distress, including feelings of rage, dread, anxiety, and discomfort. A participant even described experiencing a new and indescribable emotion when confronted with misokinesia triggers.

*"It feels like your mind is being tortured. It's like an overwhelming dread, anxiety, adrenaline, disgust. It honestly feels like a new emotion that no one can possibly describe unless you deal with it yourself."*

*"I experience a general rage when I see someone fidget even though I am very, very much not an 'angry' type of person."*

*Subtheme 1.3*: *Violent thoughts and fantasies*. A subset of participants revealed experiencing intrusive and violent thoughts, such as imagining harm to individuals responsible for triggering movements. These thoughts were recognized as irrational but showcased the intensity of some emotional responses to triggering stimuli.

*"I have thoughts that are more violent, which I would never act upon. Like for example, if I see someone tapping their fingers on a desk, my immediate thought is to chop their fingers off with a knife. This is a completely irrational response but I can't help but experience it."*

*"I start wishing with all my might that they'll stop. I do have images or 'fantasies' of that person being physically hurt by either myself or something.*

*"I get a sudden urge to slap them or hit them to make them stop."*

*"When I see someone move their foot like that, I start thinking of a way to get rid of that person in a brutal way."*

*"For example, I sadly will start hoping that they'll get into a car accident and die on their way home or someone will murder them and they'll never bother me again. I personally have never hurt anyone but the rage inside me is very real. I know it's not a natural response but I can't help it."*

*Subtheme 1.4*: *Visceral reactions*. Participants detailed intense physical reactions, such as elevated heart rate, blood pressure, and adrenaline bursts. These reactions included shaking, palpitations, sweating, and, in some instances, participants reported feeling physically ill.

*"I experience a very visceral reaction. For example, extreme discomfort, chills, and rage. My body becomes hot. My heart rate and blood pressure elevate. If the trigger lasts too long and I cannot escape it, I experience an adrenaline burst, like shaking, palpitations, sweating."*

*"When I see someone making really small repetitive movements, such as my husband bending his toes, I feel physically ill. I hold it back but I could vomit. It makes me physically sick and my heart races. Depending on how sensitive I am on the day my hands will also get clammy. My body and face become piping hot but my feet and hands are ice cold. When it's that bad, I have a hard time regulating my breathing"*

*"I start feeling a heat rush, nervousness, shortness of breath. It's a very visceral reaction. You can't really cope with it exactly"*

*"I get this heavy and intense heart palpitations, difficulty breathing, a big painful lump in my throat and tears in my eyes."*

One participant even described a unique physical sensation of intense pain in the groin area, resembling unwanted sexual arousal but with a painful, violated quality.

*"I get intense physical pain in my groin area. I know that with misophonia it can happen that people feel unwanted sexual arousal and so my misokinesia reaction is kind of similar. Except it is intensely painful. It feels like I am being sexually assaulted."*

**Subtheme 1.5**: *Distraction and avoidance*. Several participants reported attempting to self-distract, leaving the environment, and using physical barriers to block triggers from their sight or peripheral vision. Strategies varied, encompassing activities such as engaging with social media to physically obstructing or blocking the triggering movements from eyesight.

*"Ideally, I try to find ways to ignore it. However, that doesn't always work. So, if I can, I'll try to leave the space or I'll try to bury myself into my phone and scroll through social media."*

*"When I am at work trapped at a single table with several people, I have tried to block certain individuals out of my view. . . I put a large computer monitor on the table to block anyone across from me."*

*"I distract myself, or sometimes use a physical block to avoid seeing it, like holding my hand up to the side of my face to block the movement from my peripheral vision."*

*"When I see the visual chewing when I was younger, I would mimic the movement, but as I've gotten older, I now look completely away and even wear a hoodie or use my hands to prevent seeing it in my peripheral vision."*

*"I would mostly try to look away, sit in a position I won't see the movement anymore or if there is no possibility to get it out of sight otherwise, I put something, like my hand, my phone, or a book right in front of the movement so my vision is blocked."*

One participant observed that sharing a table without dividers for 8–15 hours daily posed challenges for their misokinesia triggers, but these challenges were alleviated with the shift to remote work during the COVID-19 pandemic.

*"For background, I work at a corporate job with a team ranging from 4–8 individuals at any point in time. We all sit at one large table in a room for the majority of the day. No cubicles or wall dividers—just a table. Depending on the time of year, our work day lasts anywhere from 8–15 hours. As you can imagine, being trapped in a room with several people for hours on end has presented me with several obstacles and many opportunities to develop new triggers. My work situation is literally a misokinesia nightmare. To say my productivity has taken a hit it would be an understatement. Since COVID, I have been working from home, which has been a blessing in terms of my misokinesia."*

**Subtheme 1.6**: *Coping strategies*. Some participants shared their coping strategies. One consisted of mimicry or mirroring the observed movement pattern. Another participant took a meditation-oriented approach, comprehending that restoring the balance of their "energy systems" facilitated a quicker recovery following a triggering stimulus.

*"I have had a meditation practice for many years and that does help the recovery sometimes, but I have had to do a lot of therapy for my traumas and emotional triggers including the12 step recovery work, EMDR, NLP, and a lot of energy work where I learned how to see and balance my chakras and energy system. To be honest, that latter has made the biggest difference because I can work through and process how my entire energy system is functioning when I am triggered and when it gets back into balance it also impacts my physiology and I have*

*learned what parts of my brain and body need the most attention when my misokinesia and misophonia are triggered so I know where to target and I recover much more quickly."*

*"I do think if I mimic the trigger movement it would help me depending on what it is, although this is something I have not wanted to do."*

*"Sometimes I'd even want to repeat the movement that triggers me to help relieve the stress, but this is mostly not successful at all but I still do it."*

**Theme 2: External/social impacts.** *Subtheme 2.1*: *Strain on romantic relationships.* Participants revealed challenges in maintaining long-term relationships due to the impact of MKS. Specific instances included relationship endings and difficulties related to partners' triggering behaviors. Another reported the relationship strain of even having to explain their misokinesia, as it is something others may find "weird."

*"I had a relationship end due to my misokinesia last year. . . My current boyfriend is very calm and patient. The only aspect of my misokinesia that affects our relationship is his snoring."*

*"It makes it difficult to impossible to fathom a sustainable long-term relationship. I am in my 40s and have only had 3 relationships that I consider to be long term and they were each a yearlong and long distance so I didn't have to see the person that often."*

*"My relationships are affected the most, because it's a weird thing to explain, and it won't make sense to someone who doesn't have it."*

*Subtheme 2.2*: *Struggles in family dynamics.* Misokinesia heavily influenced participants' relationships with family members, particularly when repetitive gestures were involved. Instances of strained interactions and difficulties staying present in family settings were common. Indeed, this also highlighted that MKS can be person-specific, in terms of who may or may not trigger a reaction.

*"My mother is a main trigger so it comes up a lot in our relationship when we are together and she does any repetitive gestures."*

*"The biggest aspect adversely impacted is my relationships with my family and especially my parents. They are heavily impacted because I have a very hard time being near certain family members and sometimes my parents."*

*Subtheme 2.3*: *Reluctance to communicate.* Participants expressed reluctance to communicate their condition, citing embarrassment and fear of negative reactions. This reluctance was particularly evident in interactions with acquaintances or coworkers, where potential negative consequences influenced disclosure decisions.

*"I don't really tell people anymore, it feels embarrassing. I don't tell my family about most of my visual triggers because I don't want them to feel uncomfortable by just existing in the same house as me."*

*"I have refrained from bringing up my symptoms to people who are making the movement unless I am very comfortable with that person, like my parents or a significant other. No one really outside that realm. This is because it is usually a co-worker or acquaintance who is the culprit. That person can react in two ways. Option 1 is that they apologize and make an effort*

*to stop, but this doesn't last long as the movements are habits, they don't realize they are even doing. Option 2 is that this can offend the person and they may purposefully do the movement to annoy me further. For these reasons I rarely bring up my triggers unless it is with a parent or significant other."*

*"If it is a stranger, I will ignore them."*

*Subtheme 2.4: Lack of empathy and negative reactions.* Participants shared instances where individuals around them displayed a lack of empathy, misunderstanding, or negative reactions to their misokinesia, often dismissing their condition or suggesting they could control their reactions.

*"When I tell people about it, they usually get agitated or blow it off as if I'm overreacting. I get the usual 'get over it' sort of reaction."*

*"They didn't understand and make fun of me and get angry at my anger and reactions and think I am crazy. The people whom I dated over 5 years ago hated it, but I didn't know how to communicate about it back then."*

*"Mostly they don't like how I am and think I'm not a nice person and that I should be able to control myself and that I am trying to make excuses for being irritated so much, things like —'wow what's up with you,' 'it's not that bad' or 'you're just an asshole.' I have been told to simply "get over it". I wish I could*!*"*

*"My mother reacts offended, angry, and lashes out or repeats the action more intensely."*

*Subtheme 2.5: Positive support and understanding.* Finally, despite the challenges participants faced in explaining misokinesia, some highlighted instances of positive support and understanding from friends and family members who actively learned about their triggers and offered unconditional forgiveness.

*"Others around me try to be understanding and are certainly non-judgemental, but it is a novel concept to them."*

*"My best friend and husband go out of their way to learn my triggers, avoid them, and unconditionally forgive me if I have an adverse reaction."*

*"I realized what I was dealing with so I sent articles to parents, which validated what I was feeling. They realized it was a real thing. They got more understanding, and have now stopped doing the movements."*

*"If I see leg tapping, I typically point it out to my family and ask them if they can stop and they always do."*

*"When I tell them to stop and they do it, I feel so much better."*

**Theme 3: Pragmatic factors/parameters.** Our third theme concerned the more pragmatic factors and parameters associated with episodes of MKS, as summarized in Table 3. While leg, foot, mouth, and hand movements were the most common stimuli reported to evoke MKS responses, sensitivity was expressed to a range of body-related movements, including the manual manipulation of objects (e.g., pen clicking and twirling). There was also some variability in whether individuals were sensitive to the fidgeting of babies, animals, and animate characters, whether fidgeting stimuli were more aversive if in the visual periphery vs.

**Table 3. Theme 3: The pragmatic factors/parameters associated with episodes of MKS.**

| Factor/Parameter | Response | Frequency | Percent |
|---|---|---|---|
| **Triggers** | Leg movements (e.g., bouncing, shaking, rubbing) | 18 | 20.93 |
| | Hand movements (e.g., finger tapping, pointing, rubbing) | 14 | 16.28 |
| | Foot movements (e.g., tapping, bending toes) | 13 | 15.12 |
| | Mouth movements (e.g., eating, chewing, picking teeth) | 9 | 10.47 |
| | Hair movement (e.g., plucking or playing with hair) | 8 | 9.30 |
| | Object movement (e.g., pen clicking or tapping) | 7 | 8.14 |
| | Facial movements (e.g., yawning, face touching) | 6 | 6.98 |
| | Body movements (e.g., quick, tics) | 4 | 4.65 |
| | Nail picking or biting | 4 | 4.65 |
| | Rubbing eyes or face | 2 | 2.33 |
| | Neck movements | 1 | 1.16 |
| **Bothered by Babies Fidgeting?** | Yes | 4 | 19.05 |
| | No | 17 | 80.95 |
| **Bothered by Animals Fidgeting?** | Yes | 7 | 33.33 |
| | No | 14 | 66.67 |
| **Bothered by Animated Fidgeting?** | Yes | 4 | 19.05 |
| | No | 17 | 80.95 |
| **Peripheral vs Central Vision Worse?** | Peripheral worse | 10 | 47.62 |
| | Both | 8 | 38.10 |
| | Central worse | 3 | 14.29 |
| **Worsened Over Time?** | Yes | 8 | 38.10 |
| | Same over time | 6 | 28.57 |
| | No, improved over time | 4 | 19.05 |
| | No, improved with certain triggers | 3 | 14.29 |
| **Time to Get Past It** | Few seconds to a few minutes | 10 | 47.62 |
| | As soon as I walk away | 5 | 23.81 |
| | As soon as it stops | 3 | 14.29 |
| | Still upset next day | 3 | 14.29 |
| **Anxiety Before Fidgeting** | Yes | 16 | 76.19 |
| | No | 5 | 23.81 |

occurring in central vision, and how MKS changed (if at all) with age. In terms of duration of reactions, most reported that episodes would last minutes or less once the evoking stimulus was no longer in sight; on the other hand, however, three participants did indicate that they may still feel impacts the day following an evoking event. And lastly, three fourths of the of the study participants reported that they may experience anxiety about a possible evoking event even prior to the initiation of any movement.

## Discussion

Towards trying to better-understand the subjective experiences of those with MKS, we interviewed a group of individuals self-reporting a broad range MKS. Our qualitative analysis of their responses converged on three central or overarching themes––one concerning the internal or cognitive/affective impacts of MKS, one concerning the external or social impacts, and one concerning the more pragmatic aspects of MKS episodes. But if so, how does this thematic outcome then inform on our study's set of goals?

First, one driving purpose for our study was to help inform on the possible mechanistic basis for MKS. Previously we found that MKS was not associated with either the initial visual attentional response to a kinetic stimulus [2] nor the initial affective reaction to a social-visual stimulus [3]. Given the context of these null findings, a key possibility suggested by our data here is that the mechanistic roots of MKS may not be tied to how one initially engages with an evoking stimulus, but rather, it may be linked to challenges in attentionally and/or affectively *disengaging* from a stimulus. This suggestion directly falls out of our subtheme 1.1 above, and in particular, the loss of focus many participants reported once an MKS-evoking stimulus is encountered. Indeed, one way to sum-up the subjective experiences common to the excerpts collated under this subtheme is that once a fidgeting movement was noticed, it could not be unnoticed. Importantly, this possibility––that MKS may be associated with a challenge in disengaging from an evoking stimulus––is also consistent with longstanding theories of visual attention, which functionally dissociate between mechanisms supporting attentional orienting or engagement on the one hand, and attentional disengagement on the other [e.g., 12].

In terms mechanistic insight, there are also several critical points to make regarding our third theme, the pragmatic parameters and factors associated with MKS as per Table 3. For one, MKS was not exclusively associated with either central or peripheral vision. This suggests that towards identifying possible visual-based contributors to MKS, it is not uniquely tied to either the parvocellular or magnocellular visual pathways, which are differentially weighted towards central vs. peripheral vision respectively [13, 14]. Along similar lines, not only were a full range of body parts and body movements reported to evoke MKS, but further, MKS was not just associated with human-based movement; sensitivity was also reported for animal movement and fidgeting by animated characters. If MKS does have at least a partial basis in visual-specific processing, this supports the preliminary hypothesis that the key visual areas involved may not be exclusively tied to the perception of human motion in particular [e.g., 15]. Finally, there were no systematic changes in MKS with age; some participants reported increasing sensitivity, some reported less, and others reported no apparent change. Although aging comes with predictable changes in perceptual and cognitive systems, our findings here suggest that looking to these developmental trajectories for insight into the mechanistic basis of MKS may not be fruitful.

A second goal of our study was to better understand the lived experiences of those with MKS, in the hopes of then using that information to better inform clinicians about the range of challenges it can pose. In this regard, our thematic analysis provides a compelling window into both the internal cognitive-affective correlates of an MKS episode, and the impacts of MKS on one's social relations. With respect to the former, the reactions evoked during an MKS episode go well beyond mere distractions; as we found, people reported deeply intense thoughts, feelings, and emotions. For example, words like *torture*, *overwhelming dread*, *rage*, and *disgust* to describe their experiences in the immediate moment when confronted with an MKS-evoking stimulus. Participants also described feeling *extreme discomfort*, that they *could vomit*, that they had *a hard time regulating my breathing*, and that *it feels like I am being sexually assaulted*. Likewise, these challenging visceral experiences were often accompanied by violent imagery and thoughts directed at the observed fidgeter, such as wanting to *chop off their fingers with a knife*, or *slap them or hit them*, or *get rid of them in a brutal way*. Given the intensity of these types of negative cognitive-affective experiences, it was perhaps not surprising that participants also reported a range of coping strategies to both avoid a potential fidgeter, and to help manage the course of their MKS reactions once evoked. We stress here that this constellation of MKS experiences––or the pairing of intense reactions with aggressive self-help strategies––gives stark testimony to the personal challenges faced by those with MKS.

The magnitude of these challenges is only underscored by the social impacts of MKS, as evidenced by our second overarching theme. Not only did participants report difficulties in establishing and maintaining long-term relationships, familial relations were also affected. Most notable with the latter was that for some, MKS was specifically associated with particular family members, a finding that aligns with––and further validates––personal communications two of us (SMJ and TCH) have had with other individuals with MKS who have told us about family member-specific MKS challenges. Given that the psychosocial context of the movement appears to influence responses to triggering stimuli, our focus on the social impacts of MKS aligns with the social cognition framework of misophonia. In this framework, participants may find action perception more bothersome than sound perception [16]. As our thematic analysis highlights, central to these social impacts are a concern among those with MKS about how to communicate the nature of their sensitivities to others, if at all. Participants reported feeling *embarrassed*, that often others *get agitated or blow it off as if I'm overreacting*, or that others simply don't understand (or wouldn't understand if told). However, positive benefits of communicating MKS challenges with others were also reported in our study. Namely, multiple participants indicated that speaking to others about their sensitivities did in fact lead to not just better understanding and more supportive relations, but it also helped compel others to consciously reduce their fidgeting behaviours.

Finally, the third goal of our study was to help inform on the potential development of intervention strategies for helping to alleviate the myriad challenges of MKS. Towards this end, our thematic analysis suggests that there are at least four key domains to possibly consider for intervention targeting––how one attends visually when an MKS episode may be likely, the physiological reactions that accompany an MKS episode, the thoughts and cognitions that accompany an MKS episode, and how best to disclose, discuss, and/or manage one's MKS with others. If valid, this construal of intervention domains raises important questions for future research, and most notably, whether treatment in one domain may reduce impacts in other domains. For example, if someone with MKS is treated with a $\beta$-blocker to manage their physiological reactions [17], would this reduce the intrusive negative thoughts and cognitions they may experience, and/or their need to adopt coping strategies such as blocking aversive visual inputs or avoiding social situations where fidgeting may be encountered? If mimicry is widely observed and provides relief from distress in those with misophonia [18], and given the reported mimicry in our current study of individuals with misokinesia, it suggests that these psychological phenomena may have an unconscious aspect that we have yet to explore in detail with misokinesia." Given the high prevalence of MKS in the general North American population [2] and likely elsewhere too, and given the intense nature of reactions participants reported in our current study, these become critical questions for clinicians-scientists to begin addressing.

## Limitations

In closing, it is important to note a limitation of this study: we did not receive ethics approval to record the interview sessions. Instead, the participants' responses were either typed or handwritten into a document as the interviews took place. The interviewer made sure to read the participants' responses out loud to confirm their accuracy before proceeding to the next question. Additionally, we tried to follow up with the participants 1.5 years later to ask for more details on their responses. However, only 3 out of 21 participants responded to this follow-up request, in which they provided as much detail as they felt comfortable with.

Additionally, participants were also recruited through purposive sampling, where the criteria were based on individuals self-reporting as "bothered by fidgeting." Consequently, our

sample included a significantly higher percentage of females (90%) than is usually found in psychology research. Future studies could focus on recruiting male participants with misokinesia to explore any potential gender differences in this psychological phenomenon.

## Supporting information

**S1 Table. Raw misokinesia (MkAQ), misophonia (MpAQ), and anxiety (STAI) scores reported in this manuscript.**
(DOCX)

## Author Contributions

**Conceptualization:** Sumeet M. Jaswal, Todd C. Handy.

**Data curation:** Sumeet M. Jaswal.

**Formal analysis:** Sumeet M. Jaswal, Drake Levere, Todd C. Handy.

**Funding acquisition:** Todd C. Handy.

**Investigation:** Sumeet M. Jaswal.

**Methodology:** Sumeet M. Jaswal.

**Project administration:** Sumeet M. Jaswal.

**Resources:** Sumeet M. Jaswal.

**Software:** Sumeet M. Jaswal.

**Supervision:** Todd C. Handy.

**Visualization:** Sumeet M. Jaswal.

**Writing – original draft:** Sumeet M. Jaswal, Todd C. Handy.

**Writing – review & editing:** Sumeet M. Jaswal, Todd C. Handy.

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
