## [Decision Letter · Decision Letter 0]

3 Sep 2024

PONE-D-24-27513I Struggle with Your Fidgeting: A Qualitative Study of the Personal and Social Impacts of MisokinesiaPLOS ONE

Dear Dr. Jaswal,

Thank you for submitting your manuscript to PLOS ONE. After careful consideration, we feel that it has merit but does not fully meet PLOS ONE’s publication criteria as it currently stands. Therefore, we invite you to submit a revised version of the manuscript that addresses the points raised during the review process.

This is an interesting and not much-studied condition. Please incorporate the following reviewer’s feedback and resubmit, Please also submit a letter explaning the changes made or if some of reviewer's feedback cannot be incorporated into the manuscript for any reasons,

Reviewer #1,

The authors performed a thematic analysis of interview answers based on the topic of misokinesia. This condition is a visual analog of the auditory condition, misophonia, but is highly under-studied - indeed, the authors are the only ones to publish more than case reports specifically regarding the topic. Given the scarcity of data on the condition despite its apparent prevalence, the current manuscript is very worthwhile. The writing standard is generally of a high quality (although see below for some specific changes with respect to ensuring formality) and is definitely appropriate to be published in PLoS One. I have a few slightly more major comments, in particular with respect to expanding the discussion, and then a number of minor changes.

Slightly more major:

I was intrigued as to why the question “do you have misophonia” or something similar was not explored (ideally even a misophonia questionnaire, but the question at the very least)? It would be interesting to have a full understanding of the overlap between the two conditions, given their similarity, which is something they highlighted in their 2021 study (that is, some individuals report misokinesia without misophonia). Thus, it is particularly of interest understand what the particular manifestations of these instances are from a qualitative basis. Please discuss.

Given the current study is highlighting the social nature of misokinesia, it would be relevant to at least briefly discuss a recent article on social cognition in misophonia (https://doi.org/10.1098/rstb.2023.0257), the auditory parallel condition. How do the authors feel that misokinesia may relate to social cognition, if at all?

Furthermore, given the questions and responses on mimicry, it would be appropriate to draw parallels in the discussion between the coping mechanism reported here and the nature of this response in misophonia. Indeed, another recent study has examined this in detail, highlighting the sometimes unconscious nature of the phenomenon (see https://doi.org/10.1002/jclp.23605). Please discuss with this context in mind.

Related to all three of the above comments, I feel that in general the discussion could benefit from bringing together the literature surrounding misophonia to the current study, as there is much that has already been learned there that could be transferred to and discussed in the context of understanding the mechanisms of, as well as treating, misokinesia. Given there are only 13 references at present, this seems an appropriate way of expanding this slightly and thus increasing the potential reach.

There is a very strong gender bias in the current study. Please discuss in the context of other literature (their previous studies and likely misophonia literature too) and whether this could have affected the thematic analysis.

Figure 2: there are three themes listed in the results, but only two included here. Why?

Is there no way of quantifying the degree of reporting from participants of each subtheme with the current analysis? Surely other qualitative approaches would allow for this, to indicate the strongest and weakest factors across individuals? Please at least discuss.

L173-174: “Phase 4, SMJ reviewed whether these initial themes appropriately aligned with their associated coded extracts” - this sounds very vague. How is it that you formally decided whether these themes appropriately aligned? In general, why wasn’t a more formal and unbiased approach to identifying common themes from the data not taken, such as is available with current software?

Minor comments:

L23-24 – remove “which” to make grammatical sense

L108: “AS” should say “As”

Also L108: It would be appropriate to cite the COVID-19 pandemic. One assumes that it will be a well known thing for many years to come, but it is an appropriate formality that is normal.

L109: “Zoom” should state the company details after, as is the norm with software, such as Matlab (Mathworks Inc)

L129: constructivist/interpretivist - for the more general reader, it would be appropriate to clarify what these terms mean

L154: “somewhat negatively skewed” I don’t understand the use of the word “negatively” in this context

L160: “complete” should say “completed”

L169: “one of us (SMJ)” – it would be more formal and appropriate to state “one of the authors (SMJ)”

L176: “insure” should say “ensure”

L180: “all three of us authors” – more appropriate to state “all three authors”

L185-187: “Finally, in Phase, SMJ selected the specific interview extracts to include here, SMJ and TCH co-wrote the paper, DL reviewed the paper” – this sort of text is not normally included within a manuscript, but rather in the attributions at a different section

Reviewer #2

Thank you for the opportunity to review this article. The study addresses a relatively underexplored condition, Misokinesia, which is characterized by a reduced tolerance to others' repetitive bodily movements. The research is relevant as it aims to provide foundational knowledge for understanding the personal and social impacts of Misokinesia, which could inform future empirical studies and clinical practices. he study utilizes a qualitative approach, conducting semi-structured interviews with 21 participants, which is appropriate for exploring subjective experiences and uncovering nuanced themes. The use of thematic analysis is also suitable for identifying patterns in qualitative data. The manuscript is well-structured and clearly written, with a logical flow from the introduction to the discussion. The use of direct quotes from participants enhances the richness of the data and provides a vivid illustration of the experiences described.

However, the article would need some revisions. My recommendations are as follows:

1. The sample has a majority of females which was not addressed. Is this because of higher prevalence in females as per available literature review or based on available sample collected from the group. The disproportionate distribution needs to be addressed.

2. The study mentions using STAI but no mention has been made in the results. This would be useful to check if the participants scored high for state anxiety during the interview or trait anxiety which may lead to predisposition to misokinesia and further impairment. 

3. It would be beneficial to include more detail on how potential psychological discomfort during the interviews was managed, given the sensitive nature of the topic.

4. A lot of the process of thematic analysis and identifying themes has been mentioned in results. It would be useful and easier if these are added in the methodology. It would be beneficial to mention specific findings and themes in results. 

5. The study could be more concise in the methodology, where the description of the recruitment process and data analysis could be streamlined.

Reviewer #3

Your study offers a valuable and profound understanding of the experiences of individuals with MKS, shedding light on their significant emotional and cognitive challenges. However, it has some limitations, including constraints on data collection, potential impact on data quality, sample and gender biases, and lack of generalizability. Overall, your findings provide critical insights into the experiences of individuals with MKS, emphasizing the intense and often debilitating nature of the condition. This knowledge can directly inform clinical practice, guiding more effective and empathetic care for those facing MKS.

We look forward to receiving your revised manuscript.

Kind regards,

Kamalakar Surineni, MD, MPH

Guest Editor

PLOS ONE

Journal Requirements: When submitting your revision, we need you to address these additional requirements. 1. Please ensure that your manuscript meets PLOS ONE's style requirements, including those for file naming. The PLOS ONE style templates can be found at https://journals.plos.org/plosone/s/file?id=wjVg/PLOSOne_formatting_sample_main_body.pdf and https://journals.plos.org/plosone/s/file?id=ba62/PLOSOne_formatting_sample_title_authors_affiliations.pdf 2. We note that the grant information you provided in the ‘Funding Information’ and ‘Financial Disclosure’ sections do not match.  When you resubmit, please ensure that you provide the correct grant numbers for the awards you received for your study in the ‘Funding Information’ section. 3. We note that your Data Availability Statement is currently as follows: All relevant data are within the manuscript and its Supporting Information files Please confirm at this time whether or not your submission contains all raw data required to replicate the results of your study. Authors must share the “minimal data set” for their submission. PLOS defines the minimal data set to consist of the data required to replicate all study findings reported in the article, as well as related metadata and methods (https://journals.plos.org/plosone/s/data-availability#loc-minimal-data-set-definition). For example, authors should submit the following data: - The values behind the means, standard deviations and other measures reported;- The values used to build graphs;- The points extracted from images for analysis. Authors do not need to submit their entire data set if only a portion of the data was used in the reported study. If your submission does not contain these data, please either upload them as Supporting Information files or deposit them to a stable, public repository and provide us with the relevant URLs, DOIs, or accession numbers. For a list of recommended repositories, please see https://journals.plos.org/plosone/s/recommended-repositories. If there are ethical or legal restrictions on sharing a de-identified data set, please explain them in detail (e.g., data contain potentially sensitive information, data are owned by a third-party organization, etc.) and who has imposed them (e.g., an ethics committee). Please also provide contact information for a data access committee, ethics committee, or other institutional body to which data requests may be sent. If data are owned by a third party, please indicate how others may request data access. 4. Please include your full ethics statement in the ‘Methods’ section of your manuscript file. In your statement, please include the full name of the IRB or ethics committee who approved or waived your study, as well as whether or not you obtained informed written or verbal consent. If consent was waived for your study, please include this information in your statement as well.

Additional Editor Comments:

This is an interesting and not a much-studied condition, please address the following reviewer's comments before considering for publication,

Reviewer # 1,

The authors performed a thematic analysis of interview answers based on the topic of misokinesia. This condition is a visual analog of the auditory condition, misophonia, but is highly under-studied - indeed, the authors are the only ones to publish more than case reports specifically regarding the topic. Given the scarcity of data on the condition despite its apparent prevalence, the current manuscript is very worthwhile. The writing standard is generally of a high quality (although see below for some specific changes with respect to ensuring formality) and is definitely appropriate to be published in PLoS One. I have a few slightly more major comments, in particular with respect to expanding the discussion, and then a number of minor changes.

Slightly more major:

I was intrigued as to why the question “do you have misophonia” or something similar was not explored (ideally even a misophonia questionnaire, but the question at the very least)? It would be interesting to have a full understanding of the overlap between the two conditions, given their similarity, which is something they highlighted in their 2021 study (that is, some individuals report misokinesia without misophonia). Thus, it is particularly of interest understand what the particular manifestations of these instances are from a qualitative basis. Please discuss.

Given the current study is highlighting the social nature of misokinesia, it would be relevant to at least briefly discuss a recent article on social cognition in misophonia (https://doi.org/10.1098/rstb.2023.0257), the auditory parallel condition. How do the authors feel that misokinesia may relate to social cognition, if at all?

Furthermore, given the questions and responses on mimicry, it would be appropriate to draw parallels in the discussion between the coping mechanism reported here and the nature of this response in misophonia. Indeed, another recent study has examined this in detail, highlighting the sometimes unconscious nature of the phenomenon (see https://doi.org/10.1002/jclp.23605). Please discuss with this context in mind.

Related to all three of the above comments, I feel that in general the discussion could benefit from bringing together the literature surrounding misophonia to the current study, as there is much that has already been learned there that could be transferred to and discussed in the context of understanding the mechanisms of, as well as treating, misokinesia. Given there are only 13 references at present, this seems an appropriate way of expanding this slightly and thus increasing the potential reach.

There is a very strong gender bias in the current study. Please discuss in the context of other literature (their previous studies and likely misophonia literature too) and whether this could have affected the thematic analysis.

Figure 2: there are three themes listed in the results, but only two included here. Why?

Is there no way of quantifying the degree of reporting from participants of each subtheme with the current analysis? Surely other qualitative approaches would allow for this, to indicate the strongest and weakest factors across individuals? Please at least discuss.

L173-174: “Phase 4, SMJ reviewed whether these initial themes appropriately aligned with their associated coded extracts” - this sounds very vague. How is it that you formally decided whether these themes appropriately aligned? In general, why wasn’t a more formal and unbiased approach to identifying common themes from the data not taken, such as is available with current software?

Minor comments:

L23-24 – remove “which” to make grammatical sense

L108: “AS” should say “As”

Also L108: It would be appropriate to cite the COVID-19 pandemic. One assumes that it will be a well known thing for many years to come, but it is an appropriate formality that is normal.

L109: “Zoom” should state the company details after, as is the norm with software, such as Matlab (Mathworks Inc)

L129: constructivist/interpretivist - for the more general reader, it would be appropriate to clarify what these terms mean

L154: “somewhat negatively skewed” I don’t understand the use of the word “negatively” in this context

L160: “complete” should say “completed”

L169: “one of us (SMJ)” – it would be more formal and appropriate to state “one of the authors (SMJ)”

L176: “insure” should say “ensure”

L180: “all three of us authors” – more appropriate to state “all three authors”

L185-187: “Finally, in Phase, SMJ selected the specific interview extracts to include here, SMJ and TCH co-wrote the paper, DL reviewed the paper” – this sort of text is not normally included within a manuscript, but rather in the attributions at a different section

Reviewer #2, Thank you for the opportunity to review this article. The study addresses a relatively underexplored condition, Misokinesia, which is characterized by a reduced tolerance to others' repetitive bodily movements. The research is relevant as it aims to provide foundational knowledge for understanding the personal and social impacts of Misokinesia, which could inform future empirical studies and clinical practices. he study utilizes a qualitative approach, conducting semi-structured interviews with 21 participants, which is appropriate for exploring subjective experiences and uncovering nuanced themes. The use of thematic analysis is also suitable for identifying patterns in qualitative data. The manuscript is well-structured and clearly written, with a logical flow from the introduction to the discussion. The use of direct quotes from participants enhances the richness of the data and provides a vivid illustration of the experiences described.

However, the article would need some revisions. My recommendations are as follows:

1. The sample has a majority of females which was not addressed. Is this because of higher prevalence in females as per available literature review or based on available sample collected from the group. The disproportionate distribution needs to be addressed.

2. The study mentions using STAI but no mention has been made in the results. This would be useful to check if the participants scored high for state anxiety during the interview or trait anxiety which may lead to predisposition to misokinesia and further impairment.

3. It would be beneficial to include more detail on how potential psychological discomfort during the interviews was managed, given the sensitive nature of the topic.

4. A lot of the process of thematic analysis and identifying themes has been mentioned in results. It would be useful and easier if these are added in the methodology. It would be beneficial to mention specific findings and themes in results.

5. The study could be more concise in the methodology, where the description of the recruitment process and data analysis could be streamlined.

Reviewer #3

Your study offers a valuable and profound understanding of the experiences of individuals with MKS, shedding light on their significant emotional and cognitive challenges. However, it has some limitations, including constraints on data collection, potential impact on data quality, sample and gender biases, and lack of generalizability. Overall, your findings provide critical insights into the experiences of individuals with MKS, emphasizing the intense and often debilitating nature of the condition. This knowledge can directly inform clinical practice, guiding more effective and empathetic care for those facing MKS.

Thank you,

Reviewers' comments:

Reviewer's Responses to Questions

**Comments to the Author**

1. Is the manuscript technically sound, and do the data support the conclusions?

Reviewer #1: Yes

Reviewer #2: Yes

Reviewer #3: Yes

2. Has the statistical analysis been performed appropriately and rigorously? 

Reviewer #1: N/A

Reviewer #2: N/A

Reviewer #3: Yes

3. Have the authors made all data underlying the findings in their manuscript fully available?

Reviewer #1: No

Reviewer #2: No

Reviewer #3: Yes

4. Is the manuscript presented in an intelligible fashion and written in standard English?

Reviewer #1: Yes

Reviewer #2: Yes

Reviewer #3: Yes

5. Review Comments to the Author

Reviewer #1: The authors performed a thematic analysis of interview answers based on the topic of misokinesia. This condition is a visual analog of the auditory condition, misophonia, but is highly under-studied - indeed, the authors are the only ones to publish more than case reports specifically regarding the topic. Given the scarcity of data on the condition despite its apparent prevalence, the current manuscript is very worthwhile. The writing standard is generally of a high quality (although see below for some specific changes with respect to ensuring formality) and is definitely appropriate to be published in PLoS One. I have a few slightly more major comments, in particular with respect to expanding the discussion, and then a number of minor changes.

Slightly more major:

I was intrigued as to why the question “do you have misophonia” or something similar was not explored (ideally even a misophonia questionnaire, but the question at the very least)? It would be interesting to have a full understanding of the overlap between the two conditions, given their similarity, which is something they highlighted in their 2021 study (that is, some individuals report misokinesia without misophonia). Thus, it is particularly of interest understand what the particular manifestations of these instances are from a qualitative basis. Please discuss.

Given the current study is highlighting the social nature of misokinesia, it would be relevant to at least briefly discuss a recent article on social cognition in misophonia (https://doi.org/10.1098/rstb.2023.0257), the auditory parallel condition. How do the authors feel that misokinesia may relate to social cognition, if at all?

Furthermore, given the questions and responses on mimicry, it would be appropriate to draw parallels in the discussion between the coping mechanism reported here and the nature of this response in misophonia. Indeed, another recent study has examined this in detail, highlighting the sometimes unconscious nature of the phenomenon (see https://doi.org/10.1002/jclp.23605). Please discuss with this context in mind.

Related to all three of the above comments, I feel that in general the discussion could benefit from bringing together the literature surrounding misophonia to the current study, as there is much that has already been learned there that could be transferred to and discussed in the context of understanding the mechanisms of, as well as treating, misokinesia. Given there are only 13 references at present, this seems an appropriate way of expanding this slightly and thus increasing the potential reach.

There is a very strong gender bias in the current study. Please discuss in the context of other literature (their previous studies and likely misophonia literature too) and whether this could have affected the thematic analysis.

Figure 2: there are three themes listed in the results, but only two included here. Why?

Is there no way of quantifying the degree of reporting from participants of each subtheme with the current analysis? Surely other qualitative approaches would allow for this, to indicate the strongest and weakest factors across individuals? Please at least discuss.

L173-174: “Phase 4, SMJ reviewed whether these initial themes appropriately aligned with their associated coded extracts” - this sounds very vague. How is it that you formally decided whether these themes appropriately aligned? In general, why wasn’t a more formal and unbiased approach to identifying common themes from the data not taken, such as is available with current software?

Minor comments:

L23-24 – remove “which” to make grammatical sense

L108: “AS” should say “As”

Also L108: It would be appropriate to cite the COVID-19 pandemic. One assumes that it will be a well known thing for many years to come, but it is an appropriate formality that is normal.

L109: “Zoom” should state the company details after, as is the norm with software, such as Matlab (Mathworks Inc)

L129: constructivist/interpretivist - for the more general reader, it would be appropriate to clarify what these terms mean

L154: “somewhat negatively skewed” I don’t understand the use of the word “negatively” in this context

L160: “complete” should say “completed”

L169: “one of us (SMJ)” – it would be more formal and appropriate to state “one of the authors (SMJ)”

L176: “insure” should say “ensure”

L180: “all three of us authors” – more appropriate to state “all three authors”

L185-187: “Finally, in Phase, SMJ selected the specific interview extracts to include here, SMJ and TCH co-wrote the paper, DL reviewed the paper” – this sort of text is not normally included within a manuscript, but rather in the attributions at a different section

Reviewer #2: Thank you for the opportunity to review this article. The study addresses a relatively underexplored condition, Misokinesia, which is characterized by a reduced tolerance to others' repetitive bodily movements. The research is relevant as it aims to provide foundational knowledge for understanding the personal and social impacts of Misokinesia, which could inform future empirical studies and clinical practices. he study utilizes a qualitative approach, conducting semi-structured interviews with 21 participants, which is appropriate for exploring subjective experiences and uncovering nuanced themes. The use of thematic analysis is also suitable for identifying patterns in qualitative data. The manuscript is well-structured and clearly written, with a logical flow from the introduction to the discussion. The use of direct quotes from participants enhances the richness of the data and provides a vivid illustration of the experiences described.

However, the article would need some revisions. My recommendations are as follows:

1. The sample has a majority of females which was not addressed. Is this because of higher prevalence in females as per available literature review or based on available sample collected from the group. The disproportionate distribution needs to be addressed.

2. The study mentions using STAI but no mention has been made in the results. This would be useful to check if the participants scored high for state anxiety during the interview or trait anxiety which may lead to predisposition to misokinesia and further impairment.

3. It would be beneficial to include more detail on how potential psychological discomfort during the interviews was managed, given the sensitive nature of the topic.

4. A lot of the process of thematic analysis and identifying themes has been mentioned in results. It would be useful and easier if these are added in the methodology. It would be beneficial to mention specific findings and themes in results.

5. The study could be more concise in the methodology, where the description of the recruitment process and data analysis could be streamlined.

Reviewer #3: Your study offers a valuable and profound understanding of the experiences of individuals with MKS, shedding light on their significant emotional and cognitive challenges. However, it has some limitations, including constraints on data collection, potential impact on data quality, sample and gender biases, and lack of generalizability. Overall, your findings provide critical insights into the experiences of individuals with MKS, emphasizing the intense and often debilitating nature of the condition. This knowledge can directly inform clinical practice, guiding more effective and empathetic care for those facing MKS.

6. PLOS authors have the option to publish the peer review history of their article (what does this mean?). If published, this will include your full peer review and any attached files.

Reviewer #1: No

Reviewer #2: No

Reviewer #3: **Yes: **Rajasekhar Kannali

---

## [Author Response · Author response to Decision Letter 0]

17 Oct 2024

October 14, 2024

Re: Resubmission of manuscript # PONE-D-24-27513, I Struggle with Your Fidgeting: A Qualitative Study of the Personal and Social Impacts of Misokinesia

Dear Editor: 

Thank you for the opportunity to revise and resubmit our manuscript. We appreciate that you and the reviewers both found merit in our study, and the constructive suggestions they provided for improving its clarity and import. We have made revisions in response to all of their comments, and it is our belief that the manuscript is now much improved as a result. 

Once again, thank you for considering our manuscript for publication in PLOS ONE, and the time you have invested in it.

Sincerely,

Dr. Sumeet M. Jaswal, PhD

Reviewer #1,

The authors performed a thematic analysis of interview answers based on the topic of misokinesia. This condition is a visual analog of the auditory condition, misophonia, but is highly under-studied - indeed, the authors are the only ones to publish more than case reports specifically regarding the topic. Given the scarcity of data on the condition despite its apparent prevalence, the current manuscript is very worthwhile. The writing standard is generally of a high quality (although see below for some specific changes with respect to ensuring formality) and is definitely appropriate to be published in PLoS One. I have a few slightly more major comments, in particular with respect to expanding the discussion, and then a number of minor changes.

Slightly more major:

I was intrigued as to why the question “do you have misophonia” or something similar was not explored (ideally even a misophonia questionnaire, but the question at the very least)? It would be interesting to have a full understanding of the overlap between the two conditions, given their similarity, which is something they highlighted in their 2021 study (that is, some individuals report misokinesia without misophonia). Thus, it is particularly of interest understand what the particular manifestations of these instances are from a qualitative basis. Please discuss.

On Page 7 we have added this information regarding the misophonia assessment question that we ran, and on page 12 we have added the results we obtained for this questionnaire that allows us to gather a better understanding of the degree of correlation between one’s misophonia and misokinesic sensitivities. 

Given the current study is highlighting the social nature of misokinesia, it would be relevant to at least briefly discuss a recent article on social cognition in misophonia (https://doi.org/10.1098/rstb.2023.0257), the auditory parallel condition. How do the authors feel that misokinesia may relate to social cognition, if at all?

On page 29 we added “Given that the psychosocial context of the movement appears to influence responses to triggering stimuli, our focus on the social impacts of MKS aligns with the social cognition framework of misophonia. In this framework, participants may find action perception more bothersome than sound perception (Berger, Gander & Kumar, 2024).”

Furthermore, given the questions and responses on mimicry, it would be appropriate to draw parallels in the discussion between the coping mechanism reported here and the nature of this response in misophonia. Indeed, another recent study has examined this in detail, highlighting the sometimes unconscious nature of the phenomenon (see https://doi.org/10.1002/jclp.23605). Please discuss with this context in mind.

On Page 30 we added “If mimicry is widely observed and provides relief from distress in those with misophonia (Ash et al., 2023), and given the reported mimicry in our current study of individuals with misokinesia, it suggests that these psychological phenomena may have an unconscious aspect that we have yet to explore in detail with misokinesia.”

Related to all three of the above comments, I feel that in general the discussion could benefit from bringing together the literature surrounding misophonia to the current study, as there is much that has already been learned there that could be transferred to and discussed in the context of understanding the mechanisms of, as well as treating, misokinesia. Given there are only 13 references at present, this seems an appropriate way of expanding this slightly and thus increasing the potential reach.

We do thank Reviewer 1 for their suggestions to be more inclusive of the misophonia literature to increase the reach of our work on misokinesia, and have adopted all of their suggestions above. But here we gently note that one key reason there are so few references in our paper is that there are so few papers on our actual topic -- misokinesia. Indeed, the term "misokinesia" currently only returns three hits when used as a search term in all fields on Web of Science.

There is a very strong gender bias in the current study. Please discuss in the context of other literature (their previous studies and likely misophonia literature too) and whether this could have affected the thematic analysis.

On page 31, we added “Additionally, participants were also recruited through purposive sampling, where the criteria were based on individuals self-reporting as “bothered by fidgeting." Consequently, our sample included a significantly higher percentage of females (90%) than is usually found in psychology research. Future studies could focus on recruiting male participants with misokinesia to explore any potential gender differences in this psychological phenomenon.”

Figure 2: there are three themes listed in the results, but only two included here. Why?

We believe that the third theme is best represented in a table rather than as a figure, since the practical factors and parameters concerning episodes of misokinesia sensitivity differ from the internal and external experiences expressed by our participants. Although the thematic map figure sufficiently illustrates the internal and external subthemes discussed in our document, the third theme's practical factors are more effectively conveyed alongside frequency reports, allowing readers to grasp the prevalence of these elements.

Is there no way of quantifying the degree of reporting from participants of each subtheme with the current analysis? Surely other qualitative approaches would allow for this, to indicate the strongest and weakest factors across individuals? Please at least discuss.

We chose to explore our data through thematic analyses, as we believed it offered a crucial initial examination of misokinesia in a more severe population. This study should not be viewed as conclusive; rather, it serves as a foundational step that necessitates further research. Subsequent studies could adopt a more quantitative method to investigate misokinesia and assess the impact of various factors.

L173-174: “Phase 4, SMJ reviewed whether these initial themes appropriately aligned with their associated coded extracts” - this sounds very vague. How is it that you formally decided whether these themes appropriately aligned? 

On page 9, we added further details regarding phase 4 of the qualitative thematic analysis. 

In general, why wasn’t a more formal and unbiased approach to identifying common themes from the data not taken, such as is available with current software?

We do understand that there are now AI-based alternatives for doing thematic analyses of data sets such as the one we compiled for our study. However, we chose to do a traditional thematic analysis because, quite simply, it is a formal, widely-accepted, and equally valid approach to interrogating qualitative data. 

Minor comments:

Reviewer 1 also included a number of minor comments, all of which we have adopted appropriate revisions in the manuscript. However, here we specifically address one in particular -- our decision to include author attributions in the Methods section rather than at the end of the manuscript was owing to following the checklist Standards for Reporting Qualitative Research (SRQR).

L23-24 – remove “which” to make grammatical sense

L108: “AS” should say “As”

Also L108: It would be appropriate to cite the COVID-19 pandemic. One assumes that it will be a well known thing for many years to come, but it is an appropriate formality that is normal.

L109: “Zoom” should state the company details after, as is the norm with software, such as Matlab (Mathworks Inc)

L129: constructivist/interpretivist - for the more general reader, it would be appropriate to clarify what these terms mean

L154: “somewhat negatively skewed” I don’t understand the use of the word “negatively” in this context

L160: “complete” should say “completed”

L169: “one of us (SMJ)” – it would be more formal and appropriate to state “one of the authors (SMJ)”

L176: “insure” should say “ensure”

L180: “all three of us authors” – more appropriate to state “all three authors”

L185-187: “Finally, in Phase, SMJ selected the specific interview extracts to include here, SMJ and TCH co-wrote the paper, DL reviewed the paper” – this sort of text is not normally included within a manuscript, but rather in the attributions at a different section

We agree with Reviewer 1 that such information is typically placed elsewhere in manuscripts; however, our decision to include this information in the Methods was based on following the checklist Standards for Reporting Qualitative Research (SRQR).

Reviewer #2

Thank you for the opportunity to review this article. The study addresses a relatively underexplored condition, Misokinesia, which is characterized by a reduced tolerance to others' repetitive bodily movements. The research is relevant as it aims to provide foundational knowledge for understanding the personal and social impacts of Misokinesia, which could inform future empirical studies and clinical practices. he study utilizes a qualitative approach, conducting semi-structured interviews with 21 participants, which is appropriate for exploring subjective experiences and uncovering nuanced themes. The use of thematic analysis is also suitable for identifying patterns in qualitative data. The manuscript is well-structured and clearly written, with a logical flow from the introduction to the discussion. The use of direct quotes from participants enhances the richness of the data and provides a vivid illustration of the experiences described.

However, the article would need some revisions. My recommendations are as follows:

1. The sample has a majority of females which was not addressed. Is this because of higher prevalence in females as per available literature review or based on available sample collected from the group. The disproportionate distribution needs to be addressed.

On page 31, we added “Additionally, participants were also recruited through purposive sampling, where the criteria were based on individuals self-reporting as “bothered by fidgeting." Consequently, our sample included a significantly higher percentage of females (90%) than is usually found in psychology research. Future studies could focus on recruiting male participants with misokinesia to explore any potential gender differences in this psychological phenomenon.”

2. The study mentions using STAI but no mention has been made in the results. This would be useful to check if the participants scored high for state anxiety during the interview or trait anxiety which may lead to predisposition to misokinesia and further impairment. 

On page 12 we added “We also evaluated if participants' state or trait anxiety contributed to a predisposition for misokinesia and additional impairments. Consequently, we correlated the total scores on the MkAQ with their anxiety scores (either state or trait). The findings indicated that misokinesia scores did not correlate with state anxiety (r = .308, p = .17) or with trait anxiety (r = .187, p = .429).” 

3. It would be beneficial to include more detail on how potential psychological discomfort during the interviews was managed, given the sensitive nature of the topic.

We do not have more detail to report on this issue, because our interview protocols did not include a specific management plan for possible discomfort. It was our assumption that since our participants were all self-selected for inclusion in the study, they would be comfortable discussing their experiences and challenges.

4. A lot of the process of thematic analysis and identifying themes has been mentioned in results. It would be useful and easier if these are added in the methodology. It would be beneficial to mention specific findings and themes in results. 

We moved a section of the thematic analyses description from the results to methods. 

5. The study could be more concise in the methodology, where the description of the recruitment process and data analysis could be streamlined.

While we understand Reviewer 2's comment regarding these descriptions and agree in principle that they could be streamlined, we gently note that our decision to be more expansive on these points was due to following the checklist Standards for Reporting Qualitative Research (SPQR).

Reviewer #3

Your study offers a valuable and profound understanding of the experiences of individuals with MKS, shedding light on their significant emotional and cognitive challenges. However, it has some limitations, including constraints on data collection, potential impact on data quality, sample and gender biases, and lack of generalizability. Overall, your findings provide critical insights into the experiences of individuals with MKS, emphasizing the intense and often debilitating nature of the condition. This knowledge can directly inform clinical practice, guiding more effective and empathetic care for those facing MKS.

We appreciate both Reviewer 3's positive perspective on our work, as well their identification that, like all research, our study does have its limitations. With respect to the latter, we feel that Reviewers 1 and 2 have explicitly identified key limitations, which we have addressed as noted above in responding to their comments.

---

## [Editor Report · Decision Letter 1]

21 Oct 2024

I Struggle with Your Fidgeting: A Qualitative Study of the Personal and Social Impacts of Misokinesia

PONE-D-24-27513R1

Dear Dr. Jaswal

We’re pleased to inform you that your manuscript has been judged scientifically suitable for publication and will be formally accepted for publication once it meets all outstanding technical requirements.

Kind regards,

Kamalakar Surineni, MD, MPH

Guest Editor

PLOS ONE

Additional Editor Comments (optional):

Thank you so much for positively responding to feedback and addressing the reviewer's comments.

Best, KS
---

## [Editor Report · Acceptance letter]

30 Oct 2024

PONE-D-24-27513R1 

PLOS ONE

Dear Dr. Jaswal, 

I'm pleased to inform you that your manuscript has been deemed suitable for publication in PLOS ONE. Congratulations! Your manuscript is now being handed over to our production team.

Kind regards, 

on behalf of

Dr. Kamalakar Surineni 

Guest Editor

PLOS ONE